# Excessive Oxygen Administration in High-Risk Patients Admitted to Medical and Surgical Wards Monitored by Wireless Pulse Oximeter

**DOI:** 10.3390/s24041139

**Published:** 2024-02-09

**Authors:** Clara E. Mathar, Camilla Haahr-Raunkjær, Mikkel Elvekjær, Ying Gu, Claire P. Holm, Michael P. Achiam, Lars N. Jorgensen, Eske K. Aasvang, Christian S. Meyhoff

**Affiliations:** 1Department of Anaesthesia and Intensive Care, Copenhagen University Hospital—Bispebjerg and Frederiksberg, DK-2400 Copenhagen, Denmark; clara.mathar@regionh.dk (C.E.M.); mikkelelvekjaer@gmail.com (M.E.); 2Center for Cancer and Organ Diseases, Department of Anaesthesia, Copenhagen University Hospital—Rigshospitalet, DK-2100 Copenhagen, Denmark; camilla.haahr-raunkjaer@regionh.dk (C.H.-R.); eske.kvanner.aasvang.01@regionh.dk (E.K.A.); 3Department of Health Technology, Technical University of Denmark, DK-2800 Lyngby, Denmark; yingu@dtu.dk; 4Department of Respiratory Medicine, Copenhagen University Hospital—Bispebjerg and Frederiksberg, DK-2400 Copenhagen, Denmark; claire.praest.holm@regionh.dk; 5Center for Cancer and Organ Disease, Department of Surgical Gastroenterology, Copenhagen University Hospital—Rigshospitalet, DK-2100 Copenhagen, Denmark; michael.patrick.achiam.01@regionh.dk; 6Department of Clinical Medicine, University of Copenhagen, DK-2100 Copenhagen, Denmark; lars.nannestad.joergensen@regionh.dk; 7Digestive Disease Center, Copenhagen University Hospital—Bispebjerg and Frederiksberg, DK-2400 Copenhagen, Denmark

**Keywords:** oxygen saturation, excessive oxygen administration, hyperoxemia, pulse oximeter, wireless continuous monitoring, serious adverse events

## Abstract

The monitoring of oxygen therapy when patients are admitted to medical and surgical wards could be important because exposure to excessive oxygen administration (EOA) may have fatal consequences. We aimed to investigate the association between EOA, monitored by wireless pulse oximeter, and nonfatal serious adverse events (SAEs) and mortality within 30 days. We included patients in the Capital Region of Copenhagen between 2017 and 2018. Patients were hospitalized due to acute exacerbation of chronic obstructive pulmonary disease (AECOPD) or after major elective abdominal cancer surgery, and all were treated with oxygen supply. Patients were divided into groups by their exposure to EOA: no exposure, exposure for 1–59 min or exposure over 60 min. The primary outcome was SAEs or mortality within 30 days. We retrieved data from 567 patients for a total of 43,833 h, of whom, 63% were not exposed to EOA, 26% had EOA for 1–59 min and 11% had EOA for ≥60 min. Nonfatal SAEs or mortality within 30 days developed in 24%, 12% and 22%, respectively, and the adjusted odds ratio for this was 0.98 (95% CI, 0.96–1.01) for every 10 min. increase in EOA, without any subgroup effects. In conclusion, we did not observe higher frequencies of nonfatal SAEs or mortality within 30 days in patients exposed to excessive oxygen administration.

## 1. Introduction

Supplemental oxygen therapy is one of the most prescribed treatments for patients admitted to medical and surgical wards [1]. While hypoxia can have fatal consequences, use of excessive oxygen administration (EOA) may also have detrimental side effects, such as severe complications or death, in high-risk patients [2,3,4,5]; it can occur either due to deliberate use of hyperoxia (such as use of 80% inspiratory oxygen during surgeries in attempts to increase wound oxygenation [2]) or it can occur when oxygen supplementation is given in situations without continuous SpO_2_ sensors available to titrate oxygen delivery to patient need. The prevalence of EOA is therefore assumed to be high but is also context-dependent, and therefore requires novel wireless pulse oximetry sensors to be assessed with certainty. Increasing evidence finds that hyperoxia on a cellular basis can induce reactive oxygen species that contribute to inflammation, cell death or damage [6]. Potential pulmonary side effects, besides hypercapnia, include atelectasis, pulmonary edema and respiratory failure [3]. Hyperoxia is furthermore suggested to induce coronary artery vasoconstriction, which might impair myocardial oxygenation, with an increased infarction size as a result [7]. EOA can be defined as oxygen saturation measured by pulse oximeter (SpO_2_) ≥ 99% with simultaneous supplemental oxygen therapy, unless the patient has chronic obstructive pulmonary disease (COPD) or a body mass index (BMI) ≥ 40; then, it is defined as SpO_2_ ≥ 93% with simultaneous supplemental oxygen therapy, based on peripheral saturation measured with pulse oximeter (Table 1) [8]. A large database study investigating vital signs during high-risk hospital admissions found a significantly higher 30-day mortality in patients exposed to EOA compared with adequate oxygen administration [9]. These findings regarding excessive oxygen therapy are conflicting with current guidelines from the WHO recommending intra- and postoperative hyperoxia to prevent surgical site infections [10].

Radial artery cannulation is a common procedure in the intensive care unit (ICU) for monitoring and targeting the Partial Pressure of Oxygen (PaO_2_) in patients in order to prevent EOA. However, this is not common practice in general wards, where patients usually are measured with manual pulse oximeters and deterioration can occur in the several hours between measurements. New wireless technology allows for the continuous monitoring of vital signs in all hospitalized high-risk patients outside the postanaesthetic care unit or intensive care unit. This is found to identify high frequencies of deviating vital signs, such as desaturation, that may not normally be observed in clinical rounds [11,12]. The extent of EOA can therefore also be evaluated by such technology, which is important because the magnitude and consequences of too-liberal oxygen administration in high-risk patients in general wards is sparsely elucidated, in contrast to patients admitted to ICUs [5,8,13,14].

The aim of this study was to investigate the association between EOA and subsequent complications in high-risk patients hospitalized due to acute exacerbation of chronic obstructive pulmonary disease (AECOPD) or after major elective abdominal cancer surgery. Both patient groups received supplemental oxygen and were monitored by continuous wireless pulse oximeter. The main advantage of the pulse oximeter is non-invasiveness. We hypothesized that EOA was associated with a higher frequency of nonfatal serious adverse events (SAEs) and mortality within 30 days.

## 2. Materials and Methods

### 2.1. Study Design

This was an observational study analysing data from two previous studies, ‘Wireless Assessment of Respiratory and circulatory Distress’ (WARD) observational studies [11,12] (clinicaltrials.gov identifiers: NCT03660501, NCT03491137). Both studies were designed in order to collect data on vital signs in hospitalized patients in general wards, in order to investigate deterioration and SAEs in patients that could have been prevented with wireless and continuous monitoring. Both studies were approved by the regional ethics committee (H-18026653 and H-17033535) and the Danish Data Protection Agency (2012-58-0004). All patients gave written informed consent to participation prior to inclusion. This study adheres to the STROBE guidelines.

### 2.2. Setting and Participants

Patients were enrolled in the studies mentioned above between February 2017 and July 2018 at Bispebjerg Hospital and Rigshospitalet in Copenhagen, Denmark. Eligible patients were either adults admitted to hospital with AECOPD and expecting hospitalization longer than 24 h, or patients aged ≥ 60 years undergoing major elective abdominal cancer surgery with an expected duration ≥ 2 h. Patients were excluded if not being able to cooperate, if they had allergies to plastic, plaster or silicone, had a pacemaker/ICD or if admitted only for palliative care. In total, 708 patients received oxygen supply while hospitalized and were included in this study. A total of 141 patients were excluded due to a total monitoring time of SpO_2_ less than 12 h, leaving 567 patients treated with oxygen supply and with SpO_2_ data of a minimum of 12 h; 464 were classified as surgery patients and 103 as COPD patients. The two different patient categories gave the opportunity to investigate exposure to excessive oxygen administration from different viewpoints, as it has been known for decades that COPD patients suffer from excessive oxygen administration.

### 2.3. Monitoring

A wireless pulse oximeter was used for the monitoring of SpO_2_ until discharge or a maximum of four days (Nonin WristOx 3150, Nonin Medical Inc., Minneapolis, MN, USA). SpO_2_ was measured every second, unless the patient was out of Bluetooth range or had removed the pulse oximeter in case of bathing or diagnostic imaging. Data were sent by Bluetooth to a bedside gateway and further to a hospital server with a secured wi-fi connection. Investigators’ visits included patients every day to change batteries and ensure good compliance, as the equipment had to be worn continuously. All monitored data from wireless monitoring were blinded and not visible to clinical staff. There was no intervention in patient treatment, meaning that clinical staff continued their work with clinical rounds and EWS measurements, following the daily routine.

### 2.4. Data and Variables

The primary exposure variable was the cumulative duration of EOA, as classified based on oxygen guidelines by the British Thoracic Society [8] and Troensegaard et al. [9] (Table 1).

Values are median [interquartile range 5–95%] or number (percentage).

COPD: chronic obstructive pulmonary disease. GOLD: The Global Initiative for Chronic Obstructive Lung Disease, only patients with COPD diagnosis. FEV1/FVC: Forced Expiratory Volume/Forced Vital Capacity. Location of surgery: only for surgical patients.

Secondary exposures were EOA according to two subgroups, depending on the cumulative duration of EOA (Table 1). Data on supplemental oxygen (L/min) were obtained from electronical medical records of early warning scores (EWS) [15], where the registration of any change in supplemental oxygen therapy is mandatory. We calculated the duration of EOA by evaluating COPD, BMI, SpO_2_ and supplemental oxygen therapy. A change in oxygen therapy was considered to occur at the same time it was registered in the records; likewise, if a new EWS scoring did not involve an O_2_ value, we presumed that O_2_ was unchanged and the latest registered value was used, unless 0 was registered.

Time of exposure was 12 h before SAE or death. If an SAE or fatality occurred after discharge, the exposure was the last 12 h of admission. In patients without SAEs or death, we used the first 12 h of monitoring.

We identified the number of patients receiving EOA from SpO_2_ data that exceeded the upper limits and investigated the associated SAEs. The definition of an SAE is a medical occurrence that is life-threatening, requires hospitalization or prolongation of existing hospitalization, or results in death [16].

The primary outcome was a composite of nonfatal SAEs or mortality within 30 days of the monitoring start. Secondary outcomes within 30 days were nonfatal SAEs, mortality, myocardial injury or infarction, pneumonia and respiratory failure requiring non-invasive ventilation or invasive ventilation.

### 2.5. Statistics

Data are presented as numbers (percentage) or medians and a 5–95% range for the duration of EOA. Exposure is associated with outcomes using logistic regression, with an adjustment for age, BMI ≥ 40, COPD, SpO_2_ at monitoring start and a history of myocardial infarction. The primary analysis was the cumulative duration of EOA in patients with nonfatal SAEs or mortality, versus the cumulative duration of EOA in patients without nonfatal SAEs or mortality. For statistical analyses, SAS Studio ver. 3.8 (SAS Institute, Cary, NC, USA) was used. *p* values < 0.05 was considered statistically significant.

## 3. Results

We retrieved recordings of vital data from 708 patients wearing continuous monitoring while hospitalized, whereof, 567 patients had ≥ 12 h SpO_2_ data. The cumulative duration of total SpO_2_ measurements was 43,833 h. There were 41% female patients; the median age was 71 yrs.; the median baseline O_2_ supply in all groups was 2 L/min and the highest O_2_ supply was 10 L/min. Patients were divided into three groups defined by EOA exposure: Group 1, with 358 patients (63%), were not exposed to EOA; Group 2, with 146 patients (26%), were exposed to EOA for 1–59 min; Group 3, with 63 patients (11%), were exposed to EOA ≥ 60 min; and there was a median EOA of 138 min. In Group 1, not exposed to EOA, 87% were postoperative patients, and 63% of the patients in Group 3 were admitted with AECOPD (Table 2).

There were 209 patients (38%) who were exposed to EOA, and their median duration of EOA before any SAEs was 19 min [IQR 4–78]. Development of one or more SAEs occurred in 118 patients (21%) within 30 days, and the frequencies were 24%, 12% and 22% in Groups 1–3, respectively (Table 3 and Figure 1). The adjusted odds ratio (OR) for SAEs was 0.98 [95% CI 0.95–1.01] per 10 min. exposure to EOA.

The overall 30-day mortality was 5 out of 358 patients (1.4%) in Group 1 and 2 out of 63 patients (3.2%) in Group 3; the adjusted OR for 30-day mortality was 0.99 [95% CI 0.93–1.06] per 10 min. exposure to EOA (Table 3).

The associations between 10 min. increases in EOA and the exploratory outcomes were myocardial infarction OR 1.00 [95% CI 0.94–1.07], pneumonia OR 1.00 [95% CI 0.98–1.03] or respiratory failure OR 0.99 [95% CI 0.95–1.04], requiring either NIV treatment OR 0.99 [95% CI 0.93–1.07] or invasive mechanical ventilation OR 0.99 [95% CI 0.92–1.06] (Figure 1, Table 3).

In the subgroup analysis, a total of 86 out of 464 surgery patients (19%) and 32 out of 103 patients with AECOPD (31%) experienced SAEs within 30 days (Table 3).

## 4. Discussion

When using wearable SpO_2_ sensors, this observational cohort study examined and found no significant association between EOA and subsequent SAEs or 30-day mortality in high-risk patients admitted to hospital with AECOPD or after major abdominal cancer surgery. Although the frequencies of outcomes varied according to the duration of EOA, the association with serious adverse events was not significant after adjustment for important confounders. Our findings imply that durations of oxygen supply to a higher SpO_2_ than indicated may not be strongly associated with SAEs.

A Danish cohort study by Troensegaard et al. investigated the association between 30-day mortality and inadequate, adequate and excessive oxygen administration in 11,196 patients admitted to surgical or medical wards, using point measurements from the NEWS database. Patients received a median oxygen supply of 2.2, 0.4 and 1.8 (L/min), respectively, which is comparable to our study. Significantly higher odds were found for 30-day mortality when comparing EOA to adequate oxygen, 1.46 (95% CI 1.16–1.84) [9].

A large US cohort study by McIlroy et al. [17] investigated lung, myodardial and renal injury after major surgery in 350,647 patients divided into quartiles of EOA, defined according to area under the curve of intraoperative FiO_2_ above 0.21 when the SpO_2_ was >92%. The group with the highest quartile of EOA had increases in odds for lung injury of 14% (95% CI 12–16%), in myocardial injury of 12% (95% CI 7–17%) and in acute kidney injury of 26% (95% CI 22–30%). Meyhoff et al. also investigated a high perioperative inspiratory oxygen fraction and its association with mortality in a follow-up to one of the largest trials, the PROXI trial. A total of 1386 patients underwent acute or elective abdominal surgery and were randomly assigned to receive either 80% or 30% oxygen during surgery and two hours after. After a median follow-up of 2.3 years, they found that long-term mortality was significantly increased within the high-oxygen group but only in patients who underwent cancer surgery [2].

A randomized clinical trial by Girardis et al. included 480 patients, admitted to a medical–surgical intensive care unit for an expected duration of 72 h or longer, receiving oxygen therapy to maintain an SpO_2_ of 94–98% (Conservative Group) or 97–100% (Conventional Group) when mechanically ventilated. Mortality in the Conventional Group was lower, with an absolute risk reduction of 8.6% (*p* = 0.01) [18].

These results oppose our findings, and one explanation could be that our study only had 63 patients exposed to EOA for ≥60 min. in Group 3, with a large variation of EOA durations ranging between 61 and 625 min, with a median of 2 L/min (95% CI 0–3) in oxygen supply. Our study was consequently not powered to discriminate between slightly and markedly increased oxygen tensions. Also, because of high heterogeneity in the mentioned studies due to methodological diversity, Troensegaard et al. collected point measurements from the NEWS database. A study reported problems with data quality from the early warning system caused by poor compliance by hospital staff to escalate the care and escalation protocol; this could cause inaccuracies in manually typed data, in contrast to our data from blinded and continuous monitoring [19]. Girardis et al. also included mechanical ventilated patients and targeted the SpO_2,_ as we did in our study, though the included patients were admitted to intensive care units, indicating worse physical health compared to our population group [18].

Our study did not find a statistical association in the development of myocardial injury or infarction when exposed to EOA. Some studies have assessed the cardiovascular effects of hyperoxia [20].

We did not find statistically significant associations between 10 min. increases in EOA and the subsequent occurrence of pneumonia and respiratory failure, although 22% had pneumonia in Group 3 with the longest EOA duration. This may imply that wireless monitoring of oxygen saturation provides a signal in advance before pneumonia is diagnosed. Our study sample was, however, not powered to assess pneumonia, and the findings may be due to the distribution of patients, as 63% had AECOPD as an inclusion diagnosis. This group of patients are well known to have a higher risk of pneumonia, compared to patients without a COPD diagnosis [21], which is why the results were adjusted for a number of comorbidities at the baseline. It also indicates that SpO_2_ may be more difficult to titrate to 88–92%, as indicated in patients with COPD.

Our two study populations of surgical and AECOPD patients had similar odds for an association between EOA and SAE. Baseline characteristics between the two study groups were mostly equal, but in Group 3, where 63% of the patients had AECOPD as an inclusion diagnosis, the median FEV1/FVC were lower and there were more patients with GOLD II and III, indicating worse lung status at baseline. The points estimated for pulmonary complications were higher in Group 3, but this did not remain significant after adjustment for COPD. This is not surprising, as patients with COPD are more sensitive to oxygen than other patient groups [22]. An example is a study conducted by Austin et al. comparing high-flow oxygen with titrated oxygen in 405 presumed AECOPD patients in a pre-hospital setting. They found an overall mortality of 9% vs. 4% in the high-flow group vs. the titrated group [23].

The primary strength of our study was the large amount of vital sign data from continuous and wireless monitoring for up to four days in high-risk patients, allowing us to examine SpO_2_ in an optimized hospital setting, compared with the routine infrequent EWS monitoring. We also had complete follow-up of relevant clinical outcomes within two groups of high-risk patients that are very generalizable to the population in surgical and medical wards. Certain limitations of the study are also important to emphasize.

First, SpO_2_ data were only collected in 55% and 66% of the two study populations [11,12]. Among the reasons are the following: patient compliance, resulting in the removal of the pulse oximeter when they felt restricted by the device; removal was standard during patient mobilization and personal hygiene; and technical difficulties due to battery power, a missing Bluetooth connection or turned off bedside gateway. Any adverse events in the non-recorded time unfortunately may not have been identified. Our results therefore represent durations of minimum, although we identified much higher frequencies of deviations than routine EWS monitoring [11,12]. Problems with missing data due to patient compliance are similar to other studies with continuous monitoring [24]. Furthermore, the monitoring of surgical patients began when they arrived in their respective wards and data from the period in the Post-Anesthesia Care Unit have not been included.

Second, we used a pulse oximeter to estimate the oxygen saturation, which is widely used in hospitalized patients to avoid multiple painful invasive procedures and a risk of infection. Several studies have investigated the accuracy of pulse oximeters compared to invasive arterial blood gas (SaO_2_). A meta-analysis by Jensen et al. found pulse oximeters to be accurate within 2–5% in the range of 70–100% SaO_2_ [25].

Third, patients with AECOPD in our study represent a group of medium severity, as more than half of the eligible patients were not able to give informed consent due to severe status with dyspnea. This group of patients may be at a high risk of EOA due to respiratory affection and higher need for oxygen. Our study indicates that oxygen supply is not associated with SAEs; thus, it would be interesting in future studies to include severe AECOPD patients to clarify associations between EOA and SAEs in this high-risk group.

Most out-of-hospital monitoring systems are developed through in-hospital research. Wireless SpO_2_ monitoring can be achieved in the home setting with just the same equipment as used in our study, and data can be transmitted to healthcare providers through wi-fi or 4G/5G connection. It is therefore likely that upcoming guidelines for the monitoring of oxygen saturation will apply to the in-hospital as well as the home setting, where it is documented that several deviations in SpO_2_ and other vital signs occur [26].

Future research should also investigate oxygen therapy in general wards for longer durations in a large sample size within homogenous groups, to further explore associations between EOA and individual respiratory complications and mortality. It would also be interesting to use continuous wireless monitoring to titrate oxygen therapy even more tightly in intervention studies. This would clarify the causality between exposures of no, mild and severe hyperoxia and adverse clinical outcomes.

## 5. Conclusions

Exposure to excessive oxygen administration in AECOPD and surgical patients after major abdominal cancer did not result in higher frequencies of nonfatal SAEs or mortality within 30 days. Secondary outcomes of myocardial injury and respiratory complications were not significant either.

## Figures and Tables

**Figure 1 sensors-24-01139-f001:**
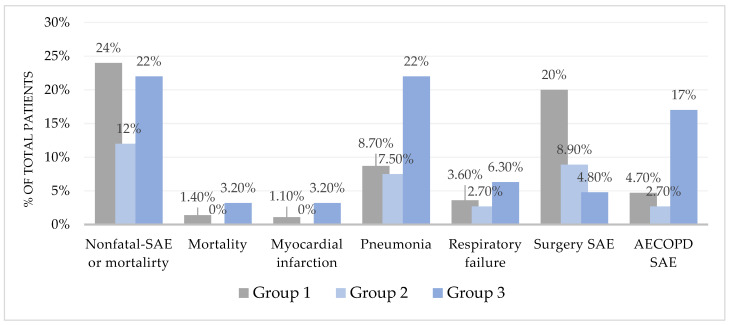
Distribution of primary and exploratory outcomes within 30 days—based on up to 96 h of continuous monitoring. Group 1: No EOA. Group 2: EOA 1–59 min. Group 3: EOA ≥ 60 min. SAE: serious adverse event. AECOPD: acute exacerbation of chronic obstructive pulmonary disease.

**Table 1 sensors-24-01139-t001:** Classification of excessive oxygen administration.

	Definition of Excessive Oxygen Administration	Cumulative Duration of Excessive Oxygen Administration
No COPD diagnosis and BMI < 40	SpO_2_ ≥ 99% with simultaneous supplemental oxygen therapy	1–59 min ≥60 min
COPD or BMI ≥ 40	SpO_2_ ≥ 93% with simultaneous supplemental oxygen therapy	1–59 min
		≥60 min

COPD: chronic obstructive pulmonary disease. BMI: body mass index. SpO_2_: saturation of peripheral oxygen.

**Table 2 sensors-24-01139-t002:** Baseline characteristics of patients receiving oxygen treatment while admitted to hospital wards.

	Group 1 No EOA (N = 358)	Group 2 EOA 1–59 min (N = 146)	Group 3 EOA ≥ 60 min (N = 63)
Age (yr)	71 [60–85]	70 [62–81]	71 [60–85]
Sex female, n (%)	133 (37%)	69 (47%)	34 (54%)
Body mass index (kg/m^2^)	25 [19–34]	25 [19–35]	25 [18–36]
Inclusion diagnosis, n (%)			
Acute exacerbation of COPD	46 (13%)	17 (12%)	40 (63%)
Major abdominal surgery	312 (87%)	129 (88%)	23 (37%)
Comorbidities, n (%)			
Cerebral disease	34 (9%)	11 (8%)	11 (17%)
Cardiac disease	76 (21%)	22 (15%)	11 (17%)
COPD	88 (25%)	30 (21%)	41 (65%)
Diabetes	58 (16%)	25 (17%)	12 (19%)
Metastasis	16 (4%)	7 (5%)	3 (5%)
Smoking status, n (%)			
Current smoker	56 (16%)	28 (19%)	14 (22%)
Previous smoker	196 (55%)	82 (56%)	37 (59%)
Never smoked	106 (30%)	36 (25%)	12 (19%)
FEV1/FVC, %	72 [42–87]	71 [42–90]	57 [37–83]
GOLD, n (%)			
GOLD I	3 (0.8%)	0	2 (3.2%)
GOLD II	16 (4.5%)	3 (2.1%)	13 (21%)
GOLD III	12 (3.4%)	7 (4.8%)	20 (32%)
GOLD IV	9 (2.5%)	6 (4.1%)	4 (6.4%)
Medical History (Charlson Comorbidity Index), n (%)			
CCI score: 2–3	49 (14%)	14 (10%)	9 (14%)
CCI score: 4–5	189 (53%)	78 (53%)	24 (38%)
CCI score: 6–7	78 (22%)	40 (27%)	18 (29%)
CCI score: 8+	32 (9%)	13 (9%)	8 (13%)
American Society of Anesthesiologists class, n (%)			
I	18 (5%)	1 (0.7%)	1 (1.6%)
II	168 (47%)	65 (45%)	10 (16%)
III	119 (33%)	61 (42%)	12 (19%)
IV	2 (0.6%)	2 (1.4%)	0
Baseline SpO_2_ (%)	96 [92–100]	98 [91–100]	96 [92–100]
Baseline O_2_ supply, L/min	2 [0–3]	2 [0–4]	2 [0–3]
Location of surgery, n (%)			
Esophagus	57 (16%)	24 (16%)	4 (6.4%)
Gastric	40 (11%)	17 (12%)	3 (4.8%)
Pancreas	114 (32%)	42 (29%)	7 (11%)
Intestines	17 (4.8%)	2 (1.4%)	3 (4.8%)
Colorectal	100 (28%)	42 (29%)	27 (43%)
Other	12 (3.6%)	10 (6.9%)	2 (3.2%)

**Table 3 sensors-24-01139-t003:** Primary and exploratory outcomes within 30 days.

	Group 1 No EOA (N = 358)	Group 2 EOA 1–59 min (N = 146)	Group 3 EOA ≥ 60 min (N = 63)	Adjusted OR	95% CI	*p*-Value
Nonfatal SAE or mortality	87 (24%)	17 (12%)	14 (22%)	0.98	[0.96–1.01]	0.24
Mortality	5 (1.4%)	0	2 (3.2%)	1.00	[0.94–1.07]	0.98
Myocardial injury or infarction	4 (1.1%)	0	2 (3.2%)	1.00	[0.95–1.08]	0.80
Pneumonia	31 (8.7%)	11 (7.5%)	14 (22%)	1.00	[0.98–1.03]	0.70
Respiratory failure	13 (3.6%)	4 (2.7%)	4 (6.3%)	0.99	[0.95–1.04]	0.77
Receiving NIV treatment	5 (1.4%)	3 (2%)	2 (3.2%)	0.99	[0.93–1.07]	0.97
Receiving invasive mechanical ventilation	7 (2%)	1 (0.7%)	1 (1.6%)	0.99	[0.93–1.06]	0.78
Subgroup analysis						
SAE in surgical patients (N = 464)	70 (22%)	13 (10%)	3 (13%)	0.97	[0.88–1.06]	0.50
SAE in patients with AECOPD (N = 103)	17 (37%)	4 (24%)	11 (28%)	0.98	[0.96–1.02]	0.35

Odds ratios are calculated per 10 min. increased exposure to EOA and adjusted for age, BMI ≥ 40, COPD, SpO_2_ at monitoring start and history of myocardial infarction. SAE: serious adverse event. NIV: non-invasive ventilation. AECOPD: acute exacerbation of chronic obstructive pulmonary disease.

## Data Availability

Data are potentially available upon request and pending internal review of such a request.

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
