# Peer review of "Excessive Oxygen Administration in High-Risk Patients Admitted to Medical and Surgical Wards Monitored by Wireless Pulse Oximeter"

_sensors, 2024, doi:10.3390/s24041139_

Round 1

Reviewer 1 Report

Comments and Suggestions for Authors

Dear authors, thanks for the opportunity to revise you paper. Authors have described retrospectively association between excessive exposure to oxygen in two different population and potential complications. The paper is well written and could be interesting for readers, but there are some concerns that should be addressed.

Major concerns:

- Authors have included and studied two very different populations: post abdominal surgical patients and acute COPD patients. It is true that they have done subgroup analysis for exposure to SAE in patients with and without EOA, but they have included all the patients together in the main analysis of outcomes and of different specific complications (such and myocardial infarction, etc) and their association to EOA. This could lead to a misinterpretation of results. 

- Do authors have obtained ethical approval for this specific retrospective study?

Minor concerns:

- Introduction. Line 46 can authors give incidence or prevalence? Line 58 reference 7 is probably not the right one, authors should have cited Troensegaard or the British thoracic society here. Lines 71-75: please state the the main advantage of pulse oximetry (wireless or not) is the practically zero invasiveness. 

- Methods: Why the age cut off in the inclusion criteria? And PMK/ICD in exclusion? Line 112-113: not scientific sentence. Please give reference. 

- Table 2: please specify that GOLD classification is for COPD patients and location of surgery is for surgical patients for clarity to the readers. 

Author Response

Please se attachted file

Reviewer 2 Report

Comments and Suggestions for Authors

In this work, the authors disclosed the association between EOA and subsequent complications in high-risk patients hospitalized due to acute exacerbation of chronic obstructive pulmonary disease (AECOPD) or after major elective abdominal cancer surgery. This topic is interesting and important to supplemental oxygen therapy. Therefore, the manuscript could be considered for publication with a major revision.

1. How to determine whether it is EOA or supplemental oxygen therapy? What is your definition of EOA?

2. What is the difference between EOA and hyperoxia?

3. It is suggested to optimize keyword on line 40.

4. It is suggested to adjust the paper format to align at both ends.

5. The writing of the paper should be improved, such as, line 75, line 223.

6. It is suggested to write the full name of the abbreviation in the article for the first time, such as SEA, PaO2, and so on.

Comments on the Quality of English Language

In this work, the authors disclosed the association between EOA and subsequent complications in high-risk patients hospitalized due to acute exacerbation of chronic obstructive pulmonary disease (AECOPD) or after major elective abdominal cancer surgery. This topic is interesting and important to supplemental oxygen therapy. Therefore, the manuscript could be considered for publication with a major revision.

1. How to determine whether it is EOA or supplemental oxygen therapy? What is your definition of EOA?

2. What is the difference between EOA and hyperoxia?

3. It is suggested to optimize keyword on line 40.

4. It is suggested to adjust the paper format to align at both ends.

5. The writing of the paper should be improved, such as, line 75, line 223.

6. It is suggested to write the full name of the abbreviation in the article for the first time, such as SEA, PaO2, and so on.

Round 2

Reviewer 1 Report

Comments and Suggestions for Authors

Authors have successfully addressed all the comments raised. Thanks again for the opportunity to revise this paper.

Reviewer 2 Report

Comments and Suggestions for Authors

In this work, the authors disclosed the association between EOA and subsequent complications in high-risk patients hospitalized due to acute exacerbation of chronic obstructive pulmonary disease (AECOPD) or after major elective abdominal cancer surgery. This topic is interesting and important to supplemental oxygen therapy. This work was well organized and detailed analysis.